# LBMamba: Locally Bi-directional Mamba

**Jingwei Zhang**[*]                                                    *jingwezhang@cs.stonybrook.edu*
*Department of Computer Science, Stony Brook University, Stony Brook, NY, USA*
**Xi Han**[*]                                                                 *xihan1@cs.stonybrook.edu*
*Department of Computer Science, Stony Brook University, Stony Brook, NY, USA*
**Hong Qin**                                                                    *qin@cs.stonybrook.edu*
*Department of Computer Science, Stony Brook University, Stony Brook, NY, USA*
**Mahdi S. Hosseini**                                              *mahdi.hosseini@concordia.ca*
*Concordia University, Montreal, Canada*
*Mila–Quebec AI Institute, Montreal, Canada*
**Dimitris Samaras**                                              *samaras@cs.stonybrook.edu*
*Department of Computer Science, Stony Brook University, Stony Brook, NY, USA*

**Reviewed on OpenReview:** *https://openreview.net/forum?id=e1aXaIXblQ*

## Abstract

Mamba, a State Space Model (SSM) that accelerates training by recasting recurrence as a parallel selective scan, has recently emerged as a linearly-scaling, efficient alternative to self-attention. Because of its unidirectional nature, each state in Mamba only has information of its previous states and is blind to states after. Current Mamba-based computer-vision methods typically overcome this limitation by augmenting Mamba's global forward scan with a global backward scan, forming a bi-directional scan that restores a full receptive field. However, this operation doubles the computational load, eroding much of the efficiency advantage that originally Mamba have. To eliminate this extra scans, we introduce **LBMamba**, a locally bi-directional SSM block that embeds a lightweight locally backward scan inside the forward selective scan and executes it entirely in per-thread registers. Building on LBMamba, we present **LBVim**, a scalable vision backbone that alternates scan directions every two layers to recover a global receptive field without extra backward sweeps. We validate the versatility of our approach on both natural images and whole slide images (WSIs). We show that our LBVim constantly offers a superior performance–throughput trade-off. That is under the same throughput, LBVim achieves 0.8% to 1.6% higher top-1 accuracy on the ImageNet-1K classification dataset, 0.6% to 2.7 % higher mIoU on the ADE20K semantic segmentation dataset, 0.9% higher $AP^b$ and 1.1% higher $AP^m$ on the COCO detection dataset. Our method serves as a general-purpose enhancement, boosting the accuracy of four SOTA Mamba models, namely VMamba, LocalVim, PlainMamba and Adventurer, by 0.5% to 3.4%. We also integrate LBMamba into the SOTA pathology multiple instance learning (MIL) approach, MambaMIL, which uses single directional scan. Experiments on 3 public WSI classification datasets show that our method achieves a relative improvement of up to 3.06% better AUC, 3.39% better F1, 1.67% better accuracy. Our code is available at https://github.com/cvlab-stonybrook/LBMamba.

## 1 Introduction

State-space models (SSMs) have emerged as a compelling alternative to self-attention for sequence modeling because their hidden-state recurrence yields linear time and memory complexity with respect to sequence length (Gu et al., 2021a; Wang et al., 2022). Yet, conventional SSMs trained with naïve recurrence is still limited by slow training and inference, as they cannot leverage efficient parallelism of modern GPUs

---

[*]These authors contributed equally to this paper.

(Baron et al., 2023). Mamba (Gu & Dao, 2023) overcomes this by decoupling the state update from the hidden-to-output convolution and reformulating the computation as a selective parallel scan that runs efficiently on modern GPUs. Consequently, Mamba matches Transformer-level accuracy on long-range tasks while exhibiting far better resolution wise scaling characteristics, making it an attractive choice for both research and production systems. It was first introduced for natural-language processing and has since been adapted to computer vision (Zhu et al., 2024; Liu et al., 2024; Huang et al., 2024). Vision models built on Mamba's selective-scan kernel delivers substantial GPU speed-ups and memory savings while consistently outperforming Transformer based baselines.

Standard computer vision mamba based models scan images multiple times from different directions to enhance their performance (Zhu et al., 2024; Liu et al., 2024; Yang et al., 2024a). There are two causes of such multiple scans: the first one is to overcome the 1D nature of Mamba. Mamba treats an image as a flattened 1D sequence, so a single left-to-right pass captures only row-wise context. To recover vertical dependencies, vision pipelines typically perform an additional scan on the column-wise ordering of patches, yielding two orthogonal sweeps that together approximate 2D spatial relationship (Liu et al., 2024; Yang et al., 2024a). Several dedicated 2D Mamba/SSM methods have recently been proposed to address this structural mismatch more directly (Zhang et al., 2024; Wang et al., 2024). The second issue is the unidirectional nature of SSMs: the latent state at position $t$ is conditioned only on past positions $1 \ldots t$ (Gu & Dao, 2023). Consequently, the model is blind to information occurring after position $t$, which often leads to sub-optimal performance on vision tasks. A common solution is to add a reverse (right-to-left or bottom-to-top) pass to restore access to future tokens, producing a bi-directional scan mechanism (Zhu et al., 2024). Although this strategy reestablishes a full receptive field, each extra sweep roughly doubles the computational load, eroding much of the efficiency advantage that originally Mamba offered.

To eliminate this extra scans required bi-directional scan, we propose LBMamba, a locally bi-directional SSM, together with the LBVim framework for vision tasks. Our main contributions are summarized bellow.

- We introduce a *local backward scan* and a *locally bi-directional SSM architecture* that integrates the backward update directly into Mamba's forward scan, thereby eliminating the costly global backward scan and markedly improving computational efficiency.

- We propose a fast *hardware-aware thread-level bi-directional scanning operator* that performs the local backward scan entirely in thread-private registers, incurring *no* additional high-bandwidth memory traffic or inter-thread communication.

- We validate the speed (aka throughput)-accuracy trade-off of our architecture by implementing it on two very different domains: natural images and Giga-pixel Whole Slide Images (WSI).

We show that instead of conducting an extra backward scan, it is more beneficial to scale up the model size under a fixed latency budget. Experiments on natural images show that LBVim achieves better performance-throughput trade-off than the baselines. Under the same throughput, LBVim achieves 0.8% to 1.6% higher top-1 accuracy on the ImageNet-1K classification dataset, 0.6% to 2.7 % higher mIoU on the ADE20K semantic segmentation dataset, 0.9% higher $AP^b$ and 1.1% higher $AP^m$ on the COCO detection dataset. Applying our approach to four diverse state-of-the-art (SOTA) models: VMamba (Liu et al., 2024), LocalVim (Huang et al., 2024), PlainMamba (Yang et al., 2024a), and Adventurer (Wang et al., 2025b), improves their accuracy by 0.5% to 3.4%, demonstrating its effectiveness as a general-purpose enhancement for Mamba-based models. We also integrate our scanning approach into the SOTA pathology multiple instance learning (MIL) approach, MambaMIL (Yang et al., 2024b). Extensive experiments on 3 public datasets for WSI classification and survival analysis datasets show that our method achieves a relative improvement of up to 3.06% better AUC, 3.39% better F1, 1.67% better accuracy.

## 2 Related work

**State Space Model (SSM)**. SSM (Kalman, 1960) is an effective sequence model that represents systems evolving over time by defining hidden states and their transitions, which makes it particularly useful for

capturing dynamic temporal behavior in sequential data. Gu et al. (2021b) unified RNNs, temporal convolutions, and neural differential equations with a linear state-space layer and demonstrated the potential of SSM-based models with the HiPPO initialization. Wang et al. (2022) proposed Bi-directional Gated SSM which is able to match BERT (Devlin et al., 2019) pretraining accuracy without attention. S4 (Gu et al., 2021a) normalized the parameter matrices into a diagonal structure and offered an option to use bi-directional convolution kernel. 2D-SSM (Baron et al., 2023) adopted a 2D-SSM recursion (Kung et al., 1977) and explored scanning image in two or four directions. Similarly, S4ND (Nguyen et al., 2022) extends S4 to images and videos by applying axis-wise updates successively along each spatial direction. Overall, SSM architectures developed prior to Mamba were constrained by slow training efficiency because there lacks an efficient parallel algorithm.

**Mamba**. Mamba (Gu & Dao, 2023) proposed a selective mechanism that makes the model parameters input-dependent and thus allows remembering important states and discard less relevant ones, alleviating the forgetting in long sequences. It also introduced a hardware-aware parallel scan algorithm that drastically accelerates state computation. Hwang et al. (2024) proposed Hydra, a bi-directional Mamba model that using generalized matrix mixers and showed a better performance than BERT. Vim (Zhu et al., 2024) and VMamba (Liu et al., 2024) are the first two mamba based models in computer vision. Vim (Zhu et al., 2024) introduced a Vision Mamba block that uses two independent selective SSMs for bi-directional aggregation of information and achieves a global receipt field. VMamba (Liu et al., 2024) introduced a pyramid Mamba network with a 4-directional scan pattern to achieve a global receipt field and also enhance spatial understanding. It also proposed some solution to alleviate the instability issue in half precision training. EfficientVMamba (Pei et al., 2024) introduced skip sampling in scanning and replace the majority of VMmaba layers to Inverted Residual blocks to speed up VMamba. Similarly, MSVMamba (Shi et al., 2024) combines low resolution scales and high resolution scans to speed up VMamba. GrootVL Xiao et al. (2024) introduced a tree style scanning patten based on the minimum spanning tree, but it did not utilize the parallel scan of Mamba. Mamba-R (Wang et al., 2025a) investigated adding regiters to the Mamba network. PlainMamba (Yang et al., 2024a) used a 4-directional selective scan and adopted a more spatially continuous scan path. 2DMmaba (Zhang et al., 2024) introduced a 2D SSM and extends Mamba's parallel scan algorithm into this 2D SSM. It also use the 4-directional scan of VMamba in their natural image applications. For all of these 4-directional scans, they are all two groups of bi-directional scans applied on two different ordering of image patches. This bi-directional scan is now the standard approach in most mamba based vision models. A previous work similar to our approach is Adventurer (Wang et al., 2025b), where the sequence is also reversed every consecutive layers to remove the bi-directional scans. Adventurer enhances the global understanding by prepending an average-pooled token while our method focuses on the local understanding of the model.

**Application of Mamba in Whole Slide Images (WSI) analysis**. WSIs are usually Giga-pixel images in the pathology domain. Most slides annotated only on slide level, requiring Multi Instance Learning (MIL) methods for WSI classification. It aggregates embedded features from a WSI for slide-level representation. MIL approaches are usually based on some attention mechanism (Ilse et al., 2018; Lu et al., 2021; Li et al., 2021) and self-attention mechanism (Shao et al., 2021). Recently, S4MIL (Fillioux et al., 2023) introduced S4 model to WSI analysis, which demonstrated the effectiveness of SSM in capturing long-range dependencies, but it does not utilize the parallel scan and thus slow. Yang et al. (Yang et al., 2024b) used Mamba and a sequence reordering mechanism to reduce overfitting and achieved even better performance. M3Mamba (Zheng et al., 2025) used dynamic memory bank to overcome the catastrophic forgetting of Mamba for long-term context. Note that these SSM based MIL methods does not use bi-directional scan.

# 3 Method

We first revisit the recursions used in Mamba and existing standard bi-directional formulations. Then we present LBMamba designed for efficiency, and an associated vision mamba framework: LBVim. Finally, we introduce the low-level CUDA design of LBMamba.

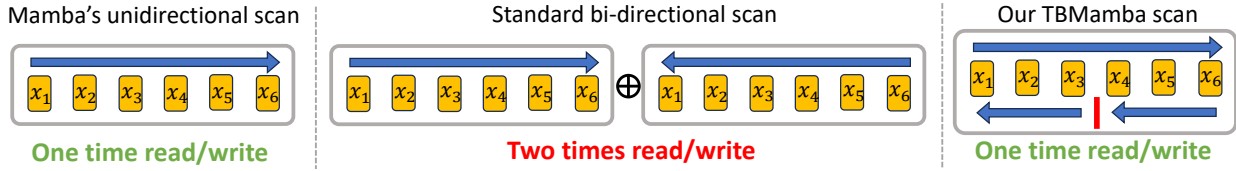

Figure 1: *(Left)*: The unidirectional scanning mechanism of the vanilla Mamba (Gu & Dao, 2023), where each state only has information of its previous/left states. *(Center)*: Standard bi-directional scanning mechanism (Zhu et al., 2024), where a dedicated backward scan is conducted and added to the forward scan. This operation involves an additional read/write of the data and thus doubles the running time. *(Right)*: Our LBMamba scan conducts a locally backward scan which is integrated into the forward scan process. This involves only one time read/write operations and thus very fast.

## 3.1 SSM in Mamba and existing standard bi-directional SSM

The original SSM in Mamba (Gu & Dao, 2023) is a mathematical model used to capture the behavior of continuous dynamic systems. To be integrated into deep models, it is discretized as:

$$h_t^f = \bar{A}^f h_{t-1}^f + \bar{B}^f x_t \tag{1}$$

$$y_t^f = C^f h_t^f + D^f x_t \tag{2}$$

where $x_t$ is the input token, $h_t^f$ is the latent state at time $t$, $y_t^f$ is the output token, $D$ is a parameter, $C^f$ is the state dimension coefficient to aggregate $N$ state dimensions into a single output and we use the superscript $f$ to denote this is a forward scan instead of a backward one. In this paper, following previous conversion (Zhu et al., 2024), "forward" and "backward" represents the scan direction rather than the forward/backward propagation in neural network training. The parameters $\bar{A}^f$, $\bar{B}^f$ and $\bar{C}^f$ are functions of the input $x_t$, which allows the SSM to dynamically adapt to the input context (known as the selective mechanism (Gu & Dao, 2023)). This aggregates important input into the hidden state while unimportant input can be ignored. Mathematically, they are:

$$\bar{A}_t^f = \exp(\Delta_t A^f) \, , \quad \bar{B}_t^f = \Delta_t B^f(x_t) \, , \quad C_t^f = C^f(x_t) \, , \quad \Delta_t = \text{softplus}(\Delta(x_t)) \tag{3}$$

where $\Delta$, $B^d$, and $C^d$ are learnable linear functions of $x_t$. $\Delta_t$ represents the time step of the discretization. The selective mechanism in the Mamba block is commonly referred to as a selective scan.

From equation 1, we can find that the output $y_t^f$ only contains information of its previous inputs $\{x_i | i < t\}$. It is also illustrated in figure 1 left. In order to enable a global recept field, previous approaches (Zhu et al., 2024; Liu et al., 2024; Yang et al., 2024a) commonly apply a backward scan and add them together as the output $y_t$ (figure 1 ceneter):

$$h_t^b = \bar{A}^b h_{t+1}^b + \bar{B}^b x_t \tag{4}$$

$$y_t^b = C^b h_t^b + D^b x_t \tag{5}$$

$$y_t = y_t^f + y_t^b \tag{6}$$

Note that the additional backward scan need to read/write the entire sequence with its parameters and thus expensive. For better distinguishing with our locally bi-direction method, we refer to this scan as *global bi-direction scan*.

## 3.2 Locally bi-directional SSM architecture

We detail the bi-directional SSM architecture, the key component of LBMamba. In contrast to the global bi-direction scan in figure 1 center that conducts a separate backward scan from the end to the beginning of the sequence, LBMamba conducts a local backward scan within sub-sequences (figure 1 right). This process can be integrated into the forward scan process and thus only requires one time read/write, which saves a lot of time.

As shown in figure 1 right, we first conduct a global forward scan as equation 1. Then we divide the input sequence of length $L$ into sub-sequences which all have a length of $M$ ($M$=3 in the figure). $M$ is set to the number of elements one thread processes and we detail it section 3.4. In each of these sub-sequences, we conduct a backward scan as that in equation 4. Specifically, the state $h_t^b$ obtained during the local backward scan is:

$$h_t^b = \begin{cases} B^f x_t & \text{if t \% M} = 0 \\ \bar{A}^f h_{t+1}^b + \bar{B}^f x_t & \text{otherwise} \end{cases} \tag{7}$$

Since this backward scan is integrated into the forward scan, we have to reuse the same parameters as the forward scan. We then add this backward hidden state and forward hidden state $h_t^f$. Because the same $B^f x_t$ is added in both forward and backward state in equation 7 and equation 1, we deduct it in the summation. This deduction does not count for computation as we can omit it by a simple programming trick (see Appendix A).

$$h_t = h_t^f + (h_t^b - B^f x_t) \,, \qquad y_t = C^f h_t + D^f x_t \tag{8}$$

## 3.3   Architecture of LBVim

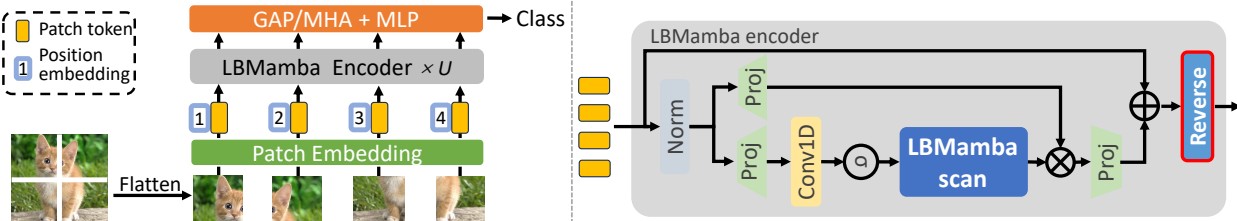

Figure 2: *(Left)*:The overall architecture of LBVim: The input image is split into patches and are embedded as patch tokens. These tokens, combined with positional embeddings, are fed to $U$ LBMamba encoders. Finally, an global average pooling (GAP) layer or a multi-head attention layer (MHA), followed by an MLP head, predicts the image class. *(Right)*: The architecture of LBVim encoder. We reverse the sequence in the end of each encoder such that the global scan direction in LBMamba is switched every two consecutive encoders, ensuring that each token achieves a global receptive field after every two encoders.

Building upon LBMamba, we introduce the overall architecture of LBVim. LBVim is based on Vim (Zhu et al., 2024). As shown in figure 2, The patch embedding and position embedding follow the same design as Vim. The embedded patches are then processed by $U$ LBMamba encoders. We replace Vim's global bi-directional selective scan with our locally bidirectional variant. As LBMamba is not a global bi-directional scan, in one encoder each token does not have a global view of the sequence. To alleviate this problem, we reverse the feature sequence at the end of each encoder to alternate the global scan direction (forward or backward). It allows each token to achieve a global receptive field after every two encoders.

Notably, we does not use the class token commonly used in (Dosovitskiy et al., 2020; Zhu et al., 2024), as it underperforms in our model. Instead, we employ an Global Average Pooling (GAP) layer followed by an MLP for final prediction in small sized models. When the size of the model scales up, we use an Multi-Head Attention with latent query mechanism (MHA) to aggregate features from the last LBMamba encoder:

$$class = MLP(softmax(\frac{qK^T(h^U)}{\sqrt{d}}V(h^U))) \tag{9}$$

where $q$ is a single learnable token, similar to the class token, $h^U$ are the output tokens from the last LBMamba encoder, $d$ is the dimension of features, $K(\cdot), V(\cdot)$ are learnable linear functions and *class* is the final prediction. Unlike the self attention mechanism, this attention only has one query and thus has linear time complexity. We find that on larger models, MHA achieves better performance than GAP.

### 3.4 Hardware-aware thread-level bi-directional scanning operator

We present our *hardware-aware scanning operator* that accelerates LBMamba scans. We first revisit the GPU storage hierarchy and then present our novel operator in detail.

**GPU storage hierarchy**. Figure 3 right illustrates the storage hierarchy of modern GPUs. The green region denotes off-chip GPU memory, with low speed and high capacity. It is referred to as *high bandwidth memory* (HBM). The orange area denotes on-chip static RAM (SRAM), with high speed but low capacity. The blue region highlights the per-thread registers, the fastest tier yet restricted to at most 255 registers per thread. In typical GPU algorithms, data is transferred from HBM to registers for computation, and the results are stored back to HBM to free SRAM and registers for succeeding computation. Because registers are private to each thread, inter-thread communication usually traverse the SRAM, which is substantially more costly than intra-thread computation. Large-scale HBM operations are even more expensive. As a consequence, many GPU algorithms (Dao et al., 2022; Dao, 2024), including Mamba, are bounded not by arithmetic computation but by memory bandwidth.

**Mamba's global forward scan.** Figure 3 outlines the vanilla Mamba's global forward scan algorithm in the blue box. Each GPU thread first fetches a tile of $P$ sequence elements from HBM into its registers ($P = 3$ in the example). The thread then performs an in-register prefix scan over these $P$ elements. We denote the partial result by $h_{i \to j}$, the hidden state obtained by scanning from time step $i$ to $j$. To extend the scan across the entire sequence, threads exchange their partial results through **SRAM**. Specifically, thread $j$ acquires a *prefix* that represents the scan of all elements preceding its own tile. For instance, thread $T_2$ in the figure receives $h_{1 \to 3}$, the scan of $x_1$ to $x_3$. Because this step requires inter-thread communication, it is significantly more expensive than intra-thread computation. After obtaining the prefix, each thread combines it with its private elements, completing the global scan, and finally writes the results back to HBM. Note that both the local scan and the prefix application traverse the same $P$ elements, doubling the arithmetic cost relative to a purely sequential scan. The conventional bi-directional scan defined by Eq. (4) is very expensive because it executes two full forward scans, thereby incurring twice the HBM traffic and inter-thread communication overhead.

**Thread level bi-directional scan.** The proposed locally bidirectional scan operator executes the backward scan entirely within each thread, avoiding any extra synchronization. As illustrated in figure 3, we begin with the same global forward scan used in vanilla Mamba. Before the results are written back to HBM, each thread performs a second scan over its private tile in the reverse (backward) direction (highlighted by the red box). This backward pass mirrors the thread scan in the forward pass but processes elements in a reversed order. The forward and backward partial sums are then added and streamed to HBM. This backward scan is thread level and does not need to apply prefix, reducing half the computation compared with a global scan. Also, because the backward scan never leaves the registers, it introduces no additional HBM traffic or inter-thread communication and is therefore extremely fast. Although this extra pass increases the overall arithmetic workload by 27%, the running time rises by only 2% (see section 4.2).

## 4 Experiments

In this section, we present a series of experiments to evaluate the performance of LBVim and compare it mainly to DeiT (Touvron et al., 2021) and Vim (Zhu et al., 2024) across various visual tasks. We also apply LBMamba to SOTA MIL method MambaMIL and SRMambaMIL to evaluate its performance on WSI datasets (Yang et al., 2024b). Following Mamba(Gu & Dao, 2023), we set the number of elements each thread thread process ($M$) based on the sequence length $L$: when $L > 256$ (images larger than $256 \times 256$), a thread processes $M = 16$ elements; for $128 < L \leq 256$ (images between $256 \times 256$ and $256 \times 128$), it processes $M = 8$ elements; and when $L \leq 128$, the workload is reduced to $M = 4$. All natural image models are trained on 4/2 Nvidia A100/H100 GPUs. All throughput, GPU memory consumption and pathology models are run on a Nvidia Quadro RTX 8000 GPU. Other implementation details are in Appendix.

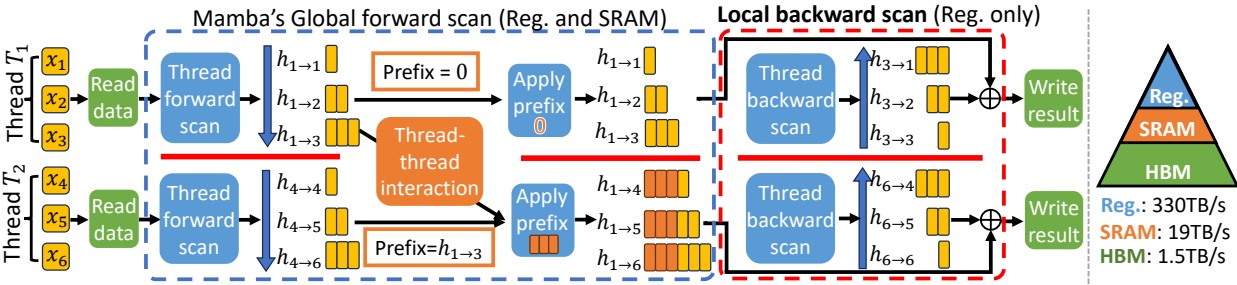

Figure 3: Our hardware-aware LBMamba CUDA operator with thread-level locally bi-directional scan. Blue color represents operations on registers (Reg.), Orange color represents operations on SRAM and green color represents those on HBM. Blue box shows the scanning operations by the *vanilla Mamba*: two threads $T_1$ and $T_2$ first loads 3 elements from HBM to registers. The global forward scan is then conducted as follows: 1) Each thread performs an in-register prefix scan over 3 elements. 2) Threads exchange their partial results through SRAM to get the prefix of each thread. 3) Each thread combines its prefix with its private elements, completing the global scan. Finally, the scanned results are write back to HBM. Red box highlights the **extra** scanning operations by the *LBMamba*: Each thread performs an in-register backward scan over 3 elements (the same as step 1 except the direction) and add it to the forward scan results. All the extra operations are in registers and thus it is very fast. $h_{i \to j}$ is the the partial result, the hidden state obtained by scanning from time step $i$ to $j$.

## 4.1 Natural image classification

We first evaluate LBVim on the ImageNet-1K dataset (Deng et al., 2009), which contains 1.28M training images and 50K validation images from 1,000 categories. All models are trained on the training set, and top-1 accuracy on the validation set is reported. For fair comparisons, we follow the training setting in (Zhu et al., 2024). To be specific, we train our models for 300 epochs using a batch size of 1,024 for tiny model and a batch size of 512 otherwise. We use the AdamW optimizer (Loshchilov & Hutter, 2017) with a momentum of 0.9, a cosine annealing learning with an initial value of $1 \times 10^{-3}$, a 5-epoch warmup period and a weight decay of 0.05. For data augmentation, we apply standard techniques: random cropping, horizontal flipping, label-smoothing regularization, mixup, and random erasing.

Table 1 shows our LBVim family with similar sized Vim baselines. At the tiny scale (192 feature dimension), with out a global backward scan, LBVim-Ti delivers a 82% higher throughput than Vim-Ti and while using 1 M fewer parameters. The top-1 accuracy trails Vim-T by 2.4%, which we attribute to the current inability of our architecture to process the class token. Comparing with Vim-Ti with global average pooling, our method achieves comparable performance (only 0.2% lower). Moving to the small setting (384 feature dimension), the accuracy gap narrows to 0.7% as the model capacity increases, yet throughput remains markedly (69%) higher.

Compared with Vim, LBVim has better speed-accuracy trade-off. We introduce two intermediate model configurations, LBVim-300 and LBVim-528. Where 300 and 528 are the dimension of features. As shown in table 1, their throughput closely match those of Vim-T and Vim-S, but surpassing the corresponding baselines by 1.6% and 0.8% in accuracy, respectively. As illustrated in figure 4, the accuracy–throughput curve of our models consistently lies in the upper-right quadrant relative to Vim, highlighting a more favorable trade-off across a wide operating range. These findings also show that our method enables a novel way to develop Mamba-based methods by scaling the model for higher accuracy, without slowing down the speed.

## 4.2 FLOPs and speed analysis

Although LBMamba performs significant additional computations compared to vanilla Mamba, it maintains fast processing speeds. At the CUDA-kernel level (table 2), LBMamba kernel executes 27% more floating-point operations than the vanilla Mamba kernel for all tested resolutions. Despite this arithmetic inflation, throughput drops by only 1.9–2.3% across all tested resolutions and the GPU memory consumption remains

Table 1: Top-1 accuracy (%) and throughput (images/second, denoted as T.P.) of LBVim variants on ImageNet-1K with $224 \times 224$ inputs. LBVim-Ti matches Vim-Ti (with global average pooling) while delivering an 82% higher throughput. LBVim-S is only 0.7 percentage points below Vim-S yet runs 69% faster. LBVim-300 and LBVim-528 attain substantially higher accuracy than Vim-Ti and Vim-S, respectively, at comparable throughput.

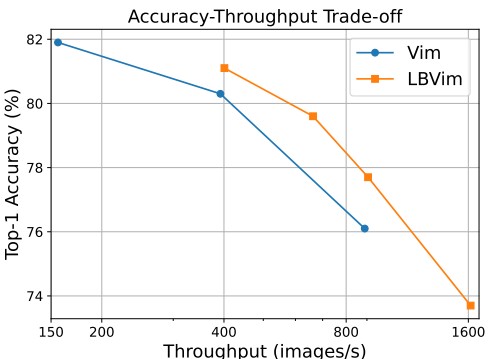

Figure 4: The accuracy-throughput trade-off curve of Vim and LBVim. The curve of LBVim consistently lies in the upper-right quadrant relative to Vim, highlighting a more favorable trade-off. We include the base version of Vim to better illustrate the trends on larger models.

| Method | #Param | FLOPs | T.P. | Top-1 acc% |
|---|---|---|---|---|
| DeiT-Ti | 6M | 1.3G | 2256 | 72.2 |
| Vim-Ti | 7M | 1.6G | 889 | 76.1 |
| Vim-Ti (GAP) | 7M | 1.6G | 897 | 73.9 |
| **LBVim-Ti** | 6M | 1.4G | 1621 | 73.7 |
| **LBVim-300** | 15M | 3.1G | 906 | **77.7** |
| DeiT-S | 22M | 4.6G | 917 | 79.8 |
| Vim-S | 26M | 5.3G | 392 | 80.3 |
| **LBVim-S** | 24M | 4.9G | 663 | 79.6 |
| **LBVim-528** | 44M | 9.0G | 398 | **81.1** |

Table 2: Comparison of floating-point operations per image (FLOPs), throughput (T.P., images or feature maps per second), and GPU memory consumption (Mem.) during inference with a batch size of 128. Input images are preloaded to GPU. For a fair comparison, we use global average pooling for Vim-Ti.

| Image size | $256 \times 256$ | | | $512 \times 512$ | | | $1024 \times 1024$ | | |
|---|---|---|---|---|---|---|---|---|---|
| Method | FLOPs | T.P. | Mem. | FLOPs | T.P. | Mem. | FLOPs | T.P. | Mem. |
| Mamba CUDA kernel | 17M | 87.4K | 209M | 69M | 14.3K | 827M | 277M | 3.8K | 3.3G |
| **LBMamba CUDA kernel** | 22M | 85.4K | 209M | 88M | 14.0K | 827M | 352M | 3.7K | 3.3G |
| DeiT-Ti | 1.7G | 1638 | 564M | 10.5G | 226 | 5.6G | 99.8G | 20 | >48G |
| Vim-Ti | 2.1G | 795 | 755M | 8.3G | 162 | 2.9G | 33.4G | 42 | 11.0G |
| **LBVim-Ti** | 1.9G | 1421 | 608M | 7.4G | 296 | 2.2G | 29.6G | 77 | 8.6G |
| **LBVim-300** | 4.1G | 799 | 862M | 16.4G | 169 | 3.4G | 65.4G | 47 | 12.6G |

unchanged. The minimal slowdown stems from the fact that the extra computations are confined to on-chip registers, incur no thread-thread interaction, and bypass the HBM or SRAM traffic. Zooming out to the model, LBVim-Ti, leveraging the LBMambakernel and omits the additional scan operation required by Vim-Ti, achieves 79%-83% higher throughput, accompanied with a 19-22% reduction in GPU memory. LBVim-300 has nearly doubled FLOPs compared with Vim-Ti but is as fast as Vim-Ti. It achieves superior performance (section 4.1) with only 14-17% additional GPU memory. This confirms that LBVim delivers a superior efficiency without sacrificing scalability. We show that this superior efficiency scale to training as well (see Appendix B)

## 4.3 Downstream tasks on natural images

We evaluate the performance of LBVim on downstream tasks, including semantic segmentation on ADE20K dataset (Zhou et al., 2019), and object detection and instance segmentation on the COCO 2017 dataset (Lin et al., 2014). The training framework is based on the MMSegmenation (Contributors, 2020) and MMDe-tection (Chen et al., 2019) libraries, following (Zhu et al., 2024) in utilizing UperNet (Xiao et al., 2018) and Cascade Mask R-CNN (Cai & Vasconcelos, 2019) as the segmentation and detection networks, respectively. For a fair comparison, we add Linear layers to LBVim-300 and LBVim-528, adjusting their output dimension to be the same as Vim-Ti and Vim-S, respectively. We find that for these two downsteam tasks, which operates on dense features and therefore does not rely on a dedicated class token, the performance gap between LBVim and the Vim baseline narrows markedly.

**Semantic Segmentation.** Table 3 left presents the mIoU results on the ADE20K dataset. The lightweight LBVim-Ti reaches 40.2 mIoU, only 0.8% below its Vim-Ti counterpart, while LBVim-S attains 44.2 mIoU, trailing Vim-S by 0.7%. When we modestly increase capacity to match Vim throughput, the gains become pronounced: LBVim-300 achieves 43.7 mIoU, 2.7 % above Vim-Ti. Similarly, LBVim-528 pushes the score to 45.5 mIoU, surpassing Vim-S by 0.6%.

**Object Detection and Instance Segmentation.** Table 3 right reports the average precision results the COCO 2017 dataset. For box AP ($AP^b$), LBVim-Ti attains 45.4%, which is on par with Vim-Ti (45.7%) and 1.1% above DeiT-Ti. Mask accuracy ($AP^m$) follows the same trend: LBVim-Ti matches Vim-Ti at 39.2% and outperforms the DeiT baseline by 1.1%. As to LBVim-300, which has similar through put as Vim-Ti, achieves 0.9% higher $AP^b$ and 1.1% higher $AP^m$. These results on segmentation and detection further demonstrate that LBVimdelivers a more favorable accuracy–efficiency trade-off on downstream tasks.

## 4.4 Ablation study

We ablate the two principal designs in LBVim—the LBMambakernel and the sequence reversing operation on the ImageNet-1K classification dataset. Table 4 shows that removing the LBMamba kernel (w.o. LBMamba) lowers accuracy by 1% with negligible impact (0.4%) on throughput. The drop confirms that the locally backward scan embedded in LBMamba improves feature propagation. On the other hand, eliminating sequence reversing operation (w.o. sequence reverse) results in a larger accuracy degradation of 4.5%. The sequence reversing operation alternates the scan direction of LBVim layers, granting each token a global receptive field every two layers, and thus strengthening long-range context modeling. Without this global receptive field, the performance drop dramatically.

We also conduct an ablation on the output aggregation scheme: global average pooling (GAP) versus multi-head attention pooling (MAP), across four capacity levels of LBVimin table 4 right. For the tiny, 300 and small variants, GAP delivers a 0.7%, 0.4% and 0.2% higher top-1 accuracy, respectively. It is clear that when the model scales up in dimension, the advantage of GAP narrows. When the model scales to LBVim-528, the MAP gains 1.1% at a negligible 0.7 % slowdown over GAP, indicating that the expressive benefit of learned pooling emerges only when sufficient parameters are available. This size-dependent trend aligns with prior observations that simple pooling is preferable for some compact models (Pan et al., 2021), whereas attention-based aggregation becomes advantageous in larger models (Dosovitskiy et al., 2020).

## 4.5 Generalizability to SOTA methods

In addition to its strong performance on the Vim baseline, LBMamba demonstrates robust generalizability when applied to other SOTA models. We integrated LBMamba with four diverse architectures: VMamba (Liu et al., 2024), LocalVim (Huang et al., 2024), PlainMamba (Yang et al., 2024a), and Adventurer (Wang

Table 3: **Left**: The performance of our LBVim on the ADE20K semantic segmentation dataset. FLOPs and and throughput (T.P.) are measured with an input size of $512 \times 2048$. **Right**: The performance of LBVim on the COCO detection dataset (image are of size $1024 \times 1024$). $AP^b$ and $AP^m$ denote the average precision for bonding boxes and masks, respectively. T.P. denotes average throughput.

| Backbone | #Param. | FLOPs | T.P. | mIoU |
|---|---|---|---|---|
| DeiT-Ti | 11M | 221G | 16 | 39.2 |
| Vim-Ti | 13M | 145G | 25 | 41.0 |
| **LBVim-Ti** | 12M | 141G | 35 | 40.2 |
| **LBVim-300** | 21M | 178G | 26 | **43.7** |
| DeiT-S | 43M | 359G | 9 | 41.0 |
| Vim-S | 46M | 227G | 15 | 44.9 |
| **LBVim-S** | 44M | 219G | 21 | 44.2 |
| **LBVim-528** | 65M | 306G | 15 | **45.5** |

| Backbone | #Param. | $AP^b$ | $AP^b_{50}$ | $AP^b_{75}$ | $AP^b_{50}$ | $AP^b_m$ | $AP^b_l$ |
|---|---|---|---|---|---|---|---|
| DeiT-Ti | 64M | 44.4 | 63.0 | 47.8 | 26.1 | 47.4 | 61.8 |
| Vim-Ti | 66M | 45.7 | 63.9 | 49.6 | 26.1 | 49.0 | 63.2 |
| **LBVim-Ti** | 66M | 45.4 | 63.8 | 49.3 | 25.5 | 49.4 | 62.4 |
| **LBVim-300** | 74M | **46.6** | **65.2** | **50.5** | **27.0** | **50.6** | **63.6** |

| Backbone | T.P. | $AP^m$ | $AP^m_{50}$ | $AP^m_{75}$ | $AP^m_{50}$ | $AP^m_m$ | $AP^m_l$ |
|---|---|---|---|---|---|---|---|
| DeiT-Ti | 7.1 | 38.1 | 59.9 | 40.5 | 18.1 | 40.5 | 58.4 |
| Vim-T | 7.3 | 39.2 | 60.9 | 41.7 | 18.2 | 41.8 | 60.2 |
| **LBVim-Ti** | 7.7 | 39.2 | 60.9 | 41.8 | 17.9 | 42.1 | 60.0 |
| **LBVim-300** | 7.3 | **40.3** | **62.5** | **43.1** | **19.3** | **43.5** | **60.4** |

Table 4: **Left:** The ablation study of LBMamba and the sequence reserving operation in LBVim on the ImageNet-1k classification dataset. **Right:** Ablation of global average pooling (GAP) vs. multi-head attention pooling (MAP) in LBVim on the ImageNet-1k classification dataset. T.P. denotes inference throughput (images/second).

| Method | T.P. | Top-1 acc% |
|---|---|---|
| LBVim-T | 1621 | **73.7** |
| w.o. LBMamba | 1628 | 72.7 |
| w.o. sequence reverse | 1711 | 69.2 |

| Model | GAP | | MAP | |
|---|---|---|---|---|
| | T.P. | Acc% | T.P | Acc% |
| LBVim-Ti | 1621 | **73.7** | 1610 | 73.0 |
| LBVim-300 | 906 | **77.7** | 901 | 77.3 |
| LBVim-S | 663 | **79.6** | 658 | 79.4 |
| LBVim-528 | 401 | 80.0 | 398 | **81.1** |

et al., 2025b), designating the enhanced versions as LBVMamba, LBLocalVim, LBPlainMamba, and LBAdventurer, respectively. For the bidirectional models (VMamba, LocalVim, and PlainMamba), we applied both LBMamba and the sequence reversal operation. For the unidirectional Adventurer, only LBMamba was applied. Each enhanced model was configured to ensure its throughput remained comparable to its respective baseline, facilitating a fair comparison. Notably, while the original LocalVim employs a search for the optimal scanning direction ("w. search"), our LBLocalVim does not. Therefore, the performance of LBLocalVim-T should be considered a lower bound. As shown in Table 5, our method consistently improves the Top-1 accuracy by 0.5% to 3.4% across all baselines, highlighting its effectiveness as a general-purpose enhancement for Mamba-based models. Detailed experimental configurations are provided in Appendix E.

Table 5: Top-1 accuracy comparison on the ImageNet-1K dataset after applying LBMamba to four SOTA baselines. Models are evaluated at comparable throughput (T.P.) to demonstrate performance gains. Adventurer is a unidirectional scan model, while the others are bidirectional. "w./w.o. search" indicates whether the LocalVim model uses its scan direction search.

| Method | #Param. | T.P. | Top-1 acc. | Method | #Param. | T.P. | Top-1 acc. |
|---|---|---|---|---|---|---|---|
| VMamba-Nano | 7M | 998 | 78.1 | PlainMamba-L1 | 7M | 385 | 77.9 |
| LBVMamba-Nano | 9M | 1003 | **79.2** | LBPlainMamba-L1 | 15M | 385 | **80.6** |
| LocalVim-T (w.o. search) | 8M | 331 | 75.8 | Adventurer | 12M | 1555 | 78.2 |
| LocalVim-T (w. search) | 8M | 331 | 76.2 | LBAdventurer | 12M | 1545 | **78.7** |
| LBLocalVim-T (w.o. search) | 15M | 356 | **79.6** | | | | |

## 4.6 WSI classification

To verify LBMamba beyond natural images, we embed it into the SOTA Multiple-Instance Learning (MIL) framework, MambaMIL and SRMambaMIL (Yang et al., 2024b), and name them LBMambaMIL and SRLBMambaMIL, respectively. We evaluate them on 3 public available Whole Slide Image datasets, PANDA (prostate grade assessment) (Bulten et al., 2022), TCGA-NSCLC (adenocarcinoma vs. squamous lung cancer) and TCGA-BRCA (breast invasive carcinoma sub-typing) (tcg). Dataset and training details are listed in Appendix E and F. As shown in table 6, with an additional locally backward scan, LBMambaMIL generally performans better than MambaMIL, achieves up to 3.06% higher accuracy, up to 3.39% higher F1 and up to 1.67% higher AUC. Similar improvement is also observed on SRMamba, SRLBMambaMIL achieves achieves up to 2.98% higher accuracy, up to 2.90% higher F1 and up to 1.48% higher AUC, demonstrating that the LBMamba is able to improve the performance of unidirectional scans. It also shows that our method generalize well on Giga-pixel pathology images.

## 4.7 Visualization of Effective Receptive Fields.

The Effective Receptive Field (ERF) (Luo et al., 2016) refers to the region in the input space that contributes to the activation of a specific output unit. We conduct a comparative analysis of the central pixel's ERF

Table 6: The comparison of accuracy (Acc), F1 and AUC on five WSI classification datasets. We conducted each experiment five times using five different random seeds and reported their mean. The highest metrics are marked as **bold**.

| Method | PANDA | | | TCGA-NSCLC | | | TCGA-BRCA | | |
|---|---|---|---|---|---|---|---|---|---|
| | *Acc* | *F1* | *AUC* | *Acc* | *F1* | *AUC* | *Acc* | *F1* | *AUC* |
| AB-MIL | 0.4883 | 0.4269 | 0.7797 | 0.8758 | 0.8756 | 0.9572 | 0.9292 | 0.8893 | 0.9747 |
| DSMIL | 0.4633 | 0.3847 | 0.7660 | 0.8782 | 0.8780 | 0.9567 | 0.9375 | 0.8961 | **0.9770** |
| CLAM | 0.4802 | 0.4224 | 0.7820 | 0.8804 | 0.8803 | 0.9536 | 0.9333 | 0.8960 | 0.9753 |
| DTFD-MIL | 0.4704 | 0.3853 | 0.7665 | 0.8736 | 0.8732 | 0.9559 | 0.9271 | 0.8809 | 0.9633 |
| TransMIL | 0.4636 | 0.3970 | 0.7728 | 0.8850 | 0.8845 | **0.9626** | **0.9375** | 0.9028 | 0.9763 |
| MambaMIL | 0.4679 | 0.4216 | 0.7781 | 0.8758 | 0.8756 | 0.9582 | 0.9333 | 0.8939 | 0.9657 |
| **LBMambaMIL** | **0.4985** | **0.4499** | **0.7948** | **0.8874** | **0.8870** | 0.9582 | 0.9333 | 0.9015 | 0.9673 |
| SRMambaMIL | 0.4711 | 0.4209 | 0.7776 | 0.8850 | 0.8849 | 0.9592 | 0.9313 | 0.8900 | 0.9657 |
| **SRLBMambaMIL** | **0.5009** | **0.4499** | **0.7924** | **0.9035** | **0.9032** | 0.9619 | **0.9375** | **0.9042** | 0.9681 |

on DeiT, Vim and LBVim at tiny and small scale, both before and after training. The results presented in figure 5 illustrate that DeiT shows global ERFs but it suffers from the quadratic complexity of self attention. Vim shows global ERFs and LBVim also shows global ERFs, proving that LBMamba with sequence reversing operation does not have a side effect on ERF.

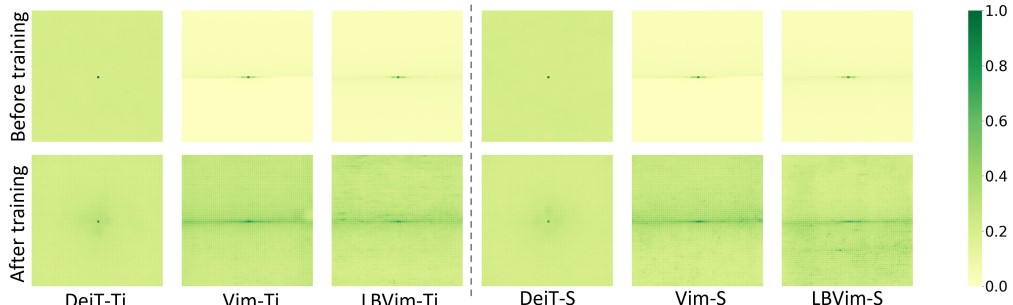

Figure 5: Comparison of Effective Receptive Fields (ERF) (Luo et al., 2016) on DeiT, Vim and LBVim at tiny and small scale. Pixels with higher intensity indicate larger responses related to the central pixel.

# 5    Conclusion

We proposed **LBMamba**, a thread-level bi-directional state-space module that marries the linear complexity of Mamba with a register-resident local backward scan. The resulting **LBVim** backbone dispenses with costly global reverse passes yet still attains a full receptive field by alternating scan directions across consecutive layers. Extensive experiments on four diverse vision tasks confirm three key findings: *Efficiency:* LBMamba adds negligible runtime overhead (2%) while eliminating one full sweep, translating to up to 83% higher throughput. *Accuracy:* Under equal or lower latency budgets, LBVim surpasses Vim by up to 1.6% ImageNet top-1 accuracy, 2.7% mIoU on ADE20K, 0.9% $AP^b$ and 1.1% $AP^m$ gains on COCO, 1.67% in AUC gains on WSI benchmarks. *Scalability:* When the size of model scales up, the accuracy–throughput Pareto front consistently dominates global bi-directional baselines, showing a better accuracy-throughput trade-off. Together, these advances indicate that local bi-directionality plus sequence reversing operation is sufficient for strong global context modeling while preserving the hallmark efficiency of SSMs.

**Limitation.** LBMamba is less effective when a dedicated class token is appended to the sequence. We systematically evaluated four common variants: head class token (prepended), middle class token (inserted at the sequence midpoint), and double class token (both prepended and appended) strategy, but none of them

achieves better performance than a simple global average pooling (see Appendix C). A plausible explanation is that the *local backward scan* treats the class token as an ordinary feature vector whose receptive field is confined to its local window; this reinforces local patterns while diluting the holistic summary that the token is meant to capture. This also limits the application of LBMamba in tokens similar to the class token, like the registers in Mamba-R (Wang et al., 2025a). Unifying local bi-directionality with an effective global summarization mechanism therefore remains an open research question, and we leave a deeper investigation of this to future work.

## Acknowledgement

This work was partially supported by USA NSF grants IIS-2123920 [D.S], IIS-2212046 [D.S], IIS-1715985 [H.Q], IIS-1812606 [H.Q], NSERC-DG RGPIN-2022-05378 [M.S.H] and Amazon Research Award [M.S.H].

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
