## A    LBVim block algorithm

Algorithm 1 shows the algorithm of LBVim block, where token sequence $\mathbf{T}_{l-1}$ is the the input and token sequence $\mathbf{T}_l^{rev}$ is the reversed output, $l$ denotes the layer number. Line 22 to 31 is our locally backward operation corresponds to equation 7 and equation 8. The simple programming trick to avoid substracting $Bx$ we mentioned in section 3.2 is on line where we do not add $Bx$ when initially when calculating $h^{backward}$ and use this result to calculate $y$. When this is finished, we then add $Bx$ back on line such that when moving to the next time step, the calculation is correct. Line 36 represents the sequence reversing operation we mentioned in section 3.3 and figure 2.

---

**Algorithm 1** LBVim Block Process

---

**Require:** token sequence $\mathbf{T}_{l-1}$ : (B, L, D)
**Ensure:** reversed token sequence $\mathbf{T}_l^{rev}$ : (B, L, D)
1: /* normalize the input sequence $\mathbf{T}'_{l-1}$ */
2: $\mathbf{T}'_{l-1}$ : (B, L, D) $\leftarrow$ **Norm**$(\mathbf{T}_{l-1})$
3: $\mathbf{x}$ : (B, L, E) $\leftarrow$ **Linear**$^{\mathbf{x}}(\mathbf{T}'_{l-1})$
4: $\mathbf{z}$ : (B, L, E) $\leftarrow$ **Linear**$^{\mathbf{z}}(\mathbf{T}'_{l-1})$
5: /* Pre-scan processes */
6: $\mathbf{x}'$ : (B, L, E) $\leftarrow$ **SiLU**$(\mathbf{Conv1d}_o(\mathbf{x}))$
7: $\mathbf{B}$ : (B, L, N) $\leftarrow$ **Linear**$^{\mathbf{B}}(\mathbf{x}'_o)$
8: $\mathbf{C}$ : (B, L, N) $\leftarrow$ **Linear**$^{\mathbf{C}}_o(\mathbf{x}')$
9: /* softplus ensures positive $\mathbf{\Delta}_o$ */
10: $\mathbf{\Delta}$ : (B, L, E) $\leftarrow \log(1 + \exp(\mathbf{Linear}^{\mathbf{\Delta}}(\mathbf{x}') + \mathbf{Parameter}^{\mathbf{\Delta}}))$
11: /* shape of $\mathbf{Parameter}^{\mathbf{A}}$ is (E, N) */
12: $\overline{\mathbf{A}}$ : (B, L, E, N) $\leftarrow \mathbf{\Delta} \bigotimes \mathbf{Parameter}^{\mathbf{A}}$
13: $\overline{\mathbf{B}}$ : (B, L, E, N) $\leftarrow \mathbf{\Delta} \bigotimes \mathbf{B}$
14: /* initialization with 0 */
15: $h^f$ : (B, E, N) $\leftarrow$ zeros (B, E, N)
16: $h^b$ : (B, E, N) $\leftarrow$ zeros (B, E, N)
17: $\mathbf{y}$ : (B, L, E) $\leftarrow$ zeros (B, L, E)
18: /* SSM forward recurrent */
19: **for** $i$ in {0, ..., L-1} **do**
20: $\quad h^f = \overline{\mathbf{A}}[:, i, :, :] \bigodot h^f + \overline{\mathbf{B}}[:, i, :, :] \bigodot \mathbf{x}'[:, i, :, \texttt{None}]$
21: **end for**
22: /* Locally backward recurrent */
23: **for** $i$ in {L-1, ..., 0} **do**
24: $\quad$ **if** $(i+1)\%M == 0$ **then**
25: $\quad\quad h^b = 0$
26: $\quad$ **else**
27: $\quad\quad h^b = \overline{\mathbf{A}_o}[:, i, :, :] \bigodot h^b$
28: $\quad$ **end if**
29: $\quad \mathbf{y}[:, i, :] = (h^f + h^b) \bigotimes \mathbf{C}[:, i, :] + \mathbf{Parameter}^{\mathbf{D}} \bigotimes x'[:, i, :]$
30: $\quad h^b = h^b + \overline{\mathbf{B}}[:, i, :, :] \bigodot \mathbf{x}'[:, i, :, \texttt{None}]$
31: **end for**
32: /* get gated $\mathbf{y}$ */
33: $\mathbf{y}'$ : (B, L, E) $\leftarrow \mathbf{y} \bigodot \mathbf{SiLU}(\mathbf{z})$
34: /* residual connection */
35: $\mathbf{T}_l$ : (B, L, D) $\leftarrow$ **Linear**$^{\mathbf{T}}(\mathbf{y}') + \mathbf{T}_{l-1}$
36: $\mathbf{T}_l^{rev}$ : (B, L, D) $\leftarrow$ **Reverse**$(\mathbf{T}_l)$
37: Return: $\mathbf{T}_l^{rev}$

---

## B    Throughput during training

Table 7 benchmarks training throughput for both the CUDA operator and the complete backbone under a batch size of 32. Similar to the findings in section 4.2, on CUDA kernel level, LBMamba is still only 2-3% slower than the vanilla Mamba kernel, showing that the thread-level bi-directional scan remains efficient

during training. Zooming to the model level, compared with Vim-Ti, LBVim-Ti remains 72-75% faster during training with nearly half of the GPU memory consumption, across all tested resolution. An interesting finding is that LBVim-300 requires 16–18% less GPU memory than Vim-Ti, given that they have similar throughput. This is probably because the standard bi-directional scan need to save more intermediate variables and thus requires more GPU memory during training. These experiments further demonstrate the superior efficiency of LBVim.

Table 7: Comparison of throughput (T.P., images or feature maps per second), and GPU memory consumption (Mem.) during training with a batch size of 32. Input images are preloaded to GPU. For a fair comparison, we use global average pooling for Vim-Ti.

| Image size | $256 \times 256$ | | $512 \times 512$ | | $1024 \times 1024$ | |
| Method | T.P. | Mem. | T.P. | Mem. | T.P. | Mem. |
|---|---|---|---|---|---|---|
| Mamba CUDA kernel | 19.1K | - | 2.4K | - | 1.0K | - |
| **LBMamba CUDA kernel** | 18.6K | - | 2.4K | - | 1.0K | - |
| Vim-Ti | 229 | 2.7G | 53 | 10.4G | 12 | 41.1G |
| **LBVim-Ti** | 394 | 1.5G | 93 | 5.5G | 21 | 21.9G |
| **LBVim-300** | 224 | 2.3G | 52 | 8.6G | 11 | 33.8G |

## C  Ablation on the class token

Table 8 shows several common strategies (Zhu et al., 2024) for incorporating a class token into LBVim-Ti on the **ImageNet-100** dataset. Consistent with our observations on ImageNet-1K (section 4.1), LBVim-Ti under performs Vim-Ti by 0.42% as it can not process the class token. Introducing a *head* class token, which is prepended to the patch sequence (ViT style (Dosovitskiy et al., 2020)), reduces accuracy by 3.22%. Adding a second *tail* token (*double* class token) offers a recovery (0.58%) but remains 2.64% lower than GAP. Placing the token in the *middle* (the strategy used by Zhu et al. (2024)) of the sequence partially alleviates the degradation, yet it still under performs the no-token design. We believe this is because the class token is treated as an ordinary feature vector, its ability to aggregate global context is diluted, leading to weaker final representations. In contrast, GAP aggregates information from all positions without introducing additional parameters or disrupting the scan pattern, making it a more compatible summarization mechanism for LBMamba-based models.

Table 8: Ablation study on common types of class token on the **ImageNet-100** dataset

| Class token type | Top-1 Acc% |
|---|---|
| Vim-Ti | 82.66 |
| LBVim-Ti (GAP) | **82.24** |
| w. head class token | 79.02 |
| w. double class token | 79.60 |
| w. middle class token | 81.04 |

## D  Implementation details of experiments on natural images classifications

The VMmaba-Nano is the same the VMamba-tiny except a 48 initial dimensionality and the number of layer for each block is 2, 2, 6, 2. LBVMamba-Nano only has two scans in every layer and we empirically scale it only by number of layers in the third block to 11, which might not be optimal. LBPlainMamba-L1 scales the 192 dimension of PlainMamba-L1 to 288. LBLocalVim-T uses only horizontal scan and vertical scan and scale the 192 dimension of LocalVim-T to 300. All the other configurations and hyper-parameters are kept the same as their corresponding baselines. Due to a GLIB_C issue, the throughput of Adventurer and LBAdventurer is measured on a different device.

# E   Implementation details of WSI experiments

**SSM models** For a fair comparison, we use a single SSM-based block with a 128-dimensional SSM and set the state dimension to 16 for all Mamba-based methods.

**Feature extractor.** We use UNI (Chen et al., 2024), a well-known and current SOTA foundation model for feature extraction, which is a ViT-L/16 pretrained on more than 100 million pathology patches from from over 100,000 H&E-stained WSIs across 20 major tissue types. UNI is pretrained in a self-supervised manner using DINOv2 (Oquab et al., 2023).

**WSI pre-processing.** We extract patches from WSIs at 20x magnification with no overlapping. The patch size is set to 512x512 pixels. We used the preprocessing tool in CLAM (Lu et al., 2021) to segment and extract tissue regions.

**Training.** We use AdamW (Loshchilov & Hutter, 2017) to optimize the models for 20 epochs of training with batch size being 1. The initial learning rate is 0.0001 and we use cosine annealing decay to adjust it.

# F   Details of WSI datasets

**Prostate cancer grading based on PANDA.** Panda dataset (Bulten et al., 2022) consists of 10,614 biopsies of prostate cancer. Its labels are their ISUP grading, totally 6 categories: *grade 0* (2890 slides), *grade 1* (2666 slides),*grade 2* (1343 slides), *grade 3* (1242 slides), *grade 4* (1249 slides), and *grade 5* (1224 slides). We random split PANDA into 80:10:10 train/validation/test sets.

**Breast invasive carcinoma subtyping on TCGA-BRCA.** TCGA-BRCA comprise 1033 H&E WSIs with 2 subtypes: *invasive ductal carcinoma* (822 slides) and *invasive lobular carcinoma* (211 slides). We follow (Chen et al., 2022) to get the train/validation/test set with 841:96:96 slides.

**Non-small cell lung carcinoma subtyping on TCGA-NSCLC.** The dataset include 957 H&E breast carcinoma WSIs, including 2 subtypes: *lung adenocarcinoma* (490 slides) and *lung squamous cell carcinoma* (468 slides). We follow (Chen et al., 2022) to split the dataset into train/validation/test set with 785:86:87 slides.

# G   Qualitative evaluation on segmentation and detection tasks

To complement the quantitative metrics of the segmentation and detection tasks presented in section 4.3, we provide a qualitative evaluation of our models on them. Figure 6 displays a qualitative comparison for semantic segmentation on the ADE20K dataset. The visualizations show that the masks produced by LBVim-Ti model are slightly less precise than those of the Vim-Ti baseline. However, the advantage of our approach becomes evident with the scaled-up LBVim-300 model. It generates visibly superior segmentation masks with cleaner boundaries and more accurate class assignments, corroborating the significant mIoU improvements reported in section 4.3. Similarly, figure 7 provides a visual comparison for object detection. In this task, LBVim-Ti achieves performance comparable to the Vim-Ti baseline, successfully identifying the main objects in the scene. Once again, the scaled-up LBVim-300 model demonstrates a clear leap in performance, producing more accurate detections. This qualitative improvement directly aligns with the higher AP scores reported in section 4.3, confirming the effectiveness of our method for detection tasks.

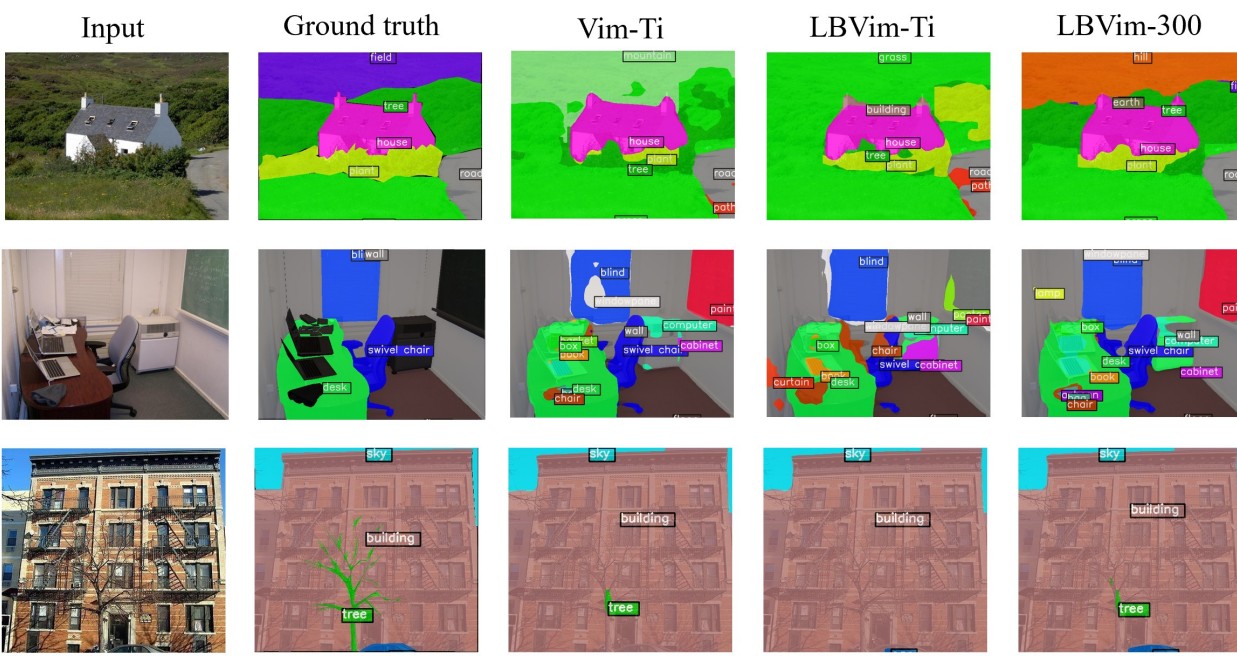

Figure 6: Qualitative segmentation comparison on the ADE20K dataset. While LBVim-Ti produces slightly less precise masks than the Vim-Ti baseline, the scaled-up LBVim-300 demonstrates the strength of our architecture with visibly superior segmentation quality.

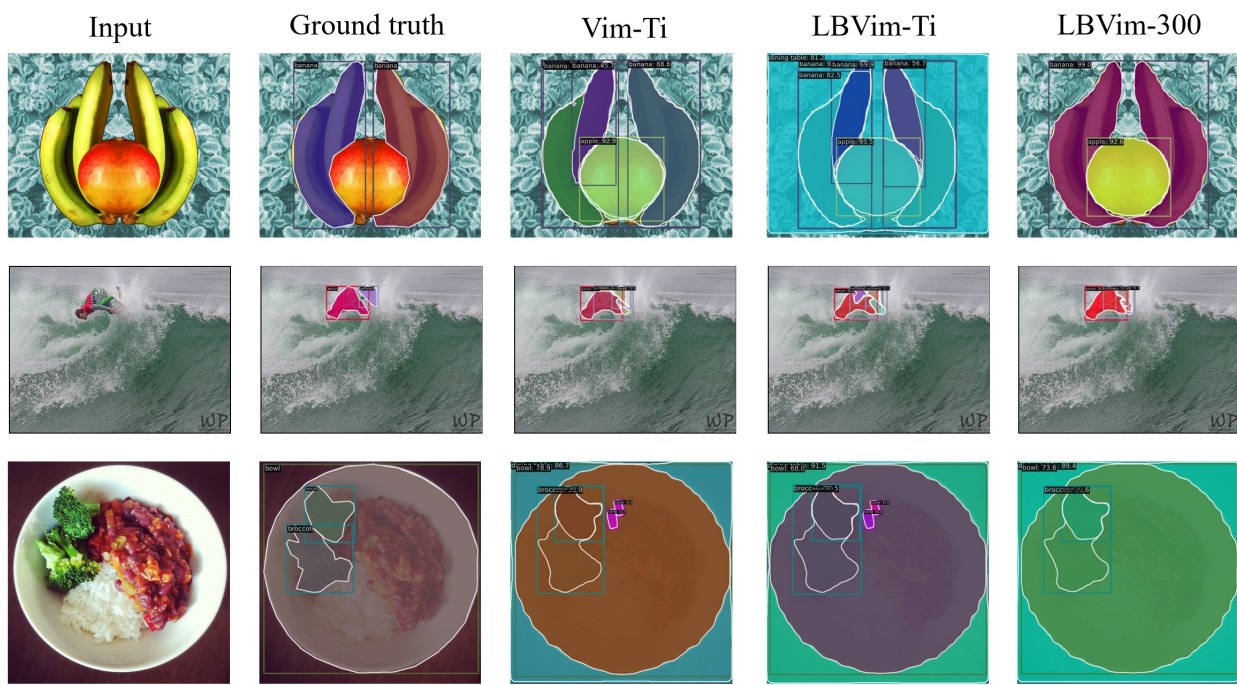

Figure 7: Qualitative comparison of detection results from Vim-Ti, LBVim-Ti, and LBVim-300 on the ADE20K dataset. LBVim-Ti achieves performance comparable to the Vim-Ti baseline, while our scaled-up LBVim-300 model produces visibly more accurate detection.