# OpenReview forum: "LBMamba: Locally Bi-directional Mamba"
_TMLR — Accepted by TMLR_

### Review · Reviewer_QtS8 · 2025-07-11

**Summary Of Contributions:**

This paper introduces LBMamba, a State Space Model (SSM) block that incorporates a lightweight, locally bi-directional scan within the standard forward selective scan mechanism. The authors claim this design, executed efficiently using per-thread registers, enhances model performance. To support this, they also propose a hardware-aware CUDA operator designed to accelerate the local backward scan.

**Audience:**

Yes

**Claims And Evidence:**

Yes

**Requested Changes:**

To be considered for publication, the paper requires a major revision to address the following points:

1.  **Strengthen Experimental Evaluation:**
    *   Conduct a comprehensive comparison against the aforementioned state-of-the-art SSMs (GrootVL, MSVMamba, Adventure, etc.) on all reported tasks.
    *   Report all relevant metrics, including FPS, for all models to ensure a fair comparison.

2.  **Justify Core Contribution:**
    *   Provide a more detailed analysis and discussion of the ablation study results. The authors must clearly demonstrate the value added by the LBMamba block itself, independent of the sequence reverse operation.
    *   Investigate and explain the performance degradation on downstream tasks.

3.  **Clarify Novelty:**
    *   Provide a clear motivation for the sequence reverse operation.
    *   Explicitly discuss the relationship and distinguish the novelty of your work from Adventure [3].

4.  **Improve Presentation:**
    *   Perform a thorough proofread of the manuscript to correct all notational inconsistencies, define all terms, and ensure consistent formatting.
    *   Add all missing citations.

**Strengths And Weaknesses:**

### Strengths

1.  **Novel Architectural Concept:** The core idea of embedding a local bi-directional scan within a single, efficient forward pass is an interesting and potentially valuable contribution to the design of SSMs.
2.  **Hardware-Aware Implementation:** The development of a custom CUDA kernel demonstrates a thoughtful approach to making the proposed architecture practical and computationally efficient.

### Weaknesses

The paper, in its current form, has several major weaknesses that limit its impact and call its contributions into question.

1.  **Insufficient Comparison with State-of-the-Art:** The experimental evaluation is not comprehensive enough to situate LBMamba within the current landscape of SSMs.
    *   **Missing Baselines:** The paper fails to discuss or compare against several highly relevant and more advanced SSM architectures, such as GrootVL [1], MSVMamba [2], and Adventure [3]. Furthermore, comparisons with variants like plainmamba, localmamba, and mamba-r [4] are necessary for a complete evaluation.
    *   **Incomplete Reporting:** Key performance metrics, such as Frames Per Second (FPS) for the DeiT baseline, are missing, which prevents a fair and thorough comparison of throughput.

2.  **Limited Empirical Performance and Ablation:** The experimental results provided are not convincing and, in some cases, undermine the paper's claims.
    *   **Marginal Gains:** The performance improvement over the Vim baseline is minimal.
    *   **Contribution Misattribution:** The ablation study in Table 4 is particularly concerning, as it suggests that the primary source of performance gain is the sequence reverse operation, not the proposed LBMamba block. This raises significant questions about the actual value of the core contribution.
    *   **Poor Downstream Performance:** The LBMamba-based model (LBVim) performs worse than the Vim baseline on object detection and segmentation tasks, indicating a potential lack of generalizability.
    *   **Kernel Efficiency:** The reported improvement from the custom CUDA kernel is described as marginal, which diminishes the impact of the hardware-related contribution.

3.  **Unclear Motivation and Novelty:**
    *   The motivation behind the sequence reverse operation is not well-explained.
    *   This operation appears to have a significant overlap with the method presented in Adventure [3], but the authors fail to discuss this, making the novelty of this component unclear.

4.  **Clarity and Presentation:** The manuscript suffers from several presentation issues that hinder readability.
    *   **Inconsistent Notation:** Symbols are used inconsistently (e.g., lowercase `q` vs. uppercase `K` in Equation 9) and are sometimes confusing (e.g., the `ft` term).
    *   **Missing Citations:** Key terms and related works are not cited properly (e.g., "MambaMIL" at the end of Section 1).

[1]. Xiao, Yicheng, et al. "Grootvl: Tree topology is all you need in state space model." NeurIPS (2024).

[2]. Shi, Yuheng, et al. "Multi-scale vmamba: Hierarchy in hierarchy visual state space model." NeurIPS (2024).

[3]. Wang, Feng, et al. "Adventurer: Optimizing Vision Mamba Architecture Designs for Efficiency." CVPR (2025).

[4]. Wang, Feng et al. “Mamba-R: Vision Mamba ALSO Needs Registers.” CVPR (2025).

---

> ### Author Response · Authors · 2025-09-01
>
> We appreciate the reviewer's valuable comment and address the issues as follows:
>
> 1. We want to emphasize that we are not proposing a new network architecture to surpass the current SOTA methods, but accelerating existing architecture by our enhanced hardware-aware algorithm. It enables scaling the existing models with much less computational overhead. We believe this work will inspire the community to investigate more GPU-level optimizations.
> Following the reviewer's recommendation, we have applied our method to plainmamba and localmamba. As shown in the table below, our method improves the performance of these two SOTA baselines by a large margin. It is important to note that the official implementation of LocalMamba does not include the code for their scan direction search. Consequently, our LBLocalVim-T operates without these optimized directions, and its result should be interpreted as a lower bound on its potential performance. Unlike our single-resolution method, MSVMamba is tailored for multiresolution applications and is thus less relevant. For experimental completeness, we apply our method to MSVMamba's single-resolution variant: VMamba. The results show a 1.1% performance improvement for VMamba. The multi-resolution version is beyond the scope of this paper and we leave it to future work. We also observed a 0.5% gain on the Adventurer baseline, which is further detailed in our response to point 3.
> Regarding the other baselines mentioned: the vision model of GrootVL is GrootV, which is Mamba-inspired but not Mamba-based. It proposes an MST-based scan path over the tokens, and employs a sequential, unparallelized scan algorithm. GrootV lacks Mamba's key component: parallel scan, and its implementation is independent of Mamba. LBMamba is strictly Mamba-based and thus applying LBMamba to GrootV is not technically feasible. Mamba-R utilizes registers in a manner analogous to class tokens. As discussed in our Limitations section, our method is not currently compatible with this architectural feature.
> We have incorporated a discussion of all these baselines into our revised manuscript and have also added FPS metrics to the DeiT comparisons for completeness.
>
> | Method | #Param. | T.P. | Top-1 acc. |
> | :--- | :---: | :---: | :---: |
> | VMamba-Nano | 7M | 998 | 78.1 |
> | LBVMamba-Nano | 9M | 1003 | **79.2** |
> | PlainMamba-L1 | 7M | 385 | 77.9 |
> | LBPlainMamba-L1 | 15M | 385 | **80.6** |
> | LocalVim-T (w.o. search) | 8M | 331 | 75.8 |
> | LocalVim-T (w. search) | 8M | 331 | 76.2 |
> | LBLocalVim-T (w.o. search) | 15M | 356 | **79.6** |
>
> 2. We would like to clarify that the sequence reverse operation is a straightforward technique to enhance our model, and we do not claim it as a novel contribution. In the context of the highly optimized ImageNet-1K benchmark, we contend that the resulting 1.0% performance improvement is not marginal.
> Regarding downstream performance, we would like to clarify that, similar to the classification tasks, on object detection and segmentation tasks, Vim should be compared with LBVim under similar throughput. In Table 3, the performance of LBVim-Ti is indeed lower than Vim-Ti but the performance of LBVim-300, with a comparable throughput to Vim-Ti, is much higher than Vim-Ti, showcasing our method's superior performance-throughput trade-off.
>
> 3. We would like to clarify that the motivation for incorporating the sequence reverse operation is to provide each token with a global receptive field, compensating for the inherently local receptive field of LBMamba. This operation is straightforward. As we discussed in the related work section in the revised submission, our motivation aligns with that of Adventurer, which also utilizes this straightforward sequence reversal. Adventurer enhances global understanding by prepending an average-pooled token, whereas LBMamba focuses on the local understanding of the model, and thus, we think these two methods are orthogonal. To validate this orthogonality, we integrated LBMamba with Adventurer and observed a further performance increase of 0.5% with negligible throughput overhead, suggesting their complementary nature.
>
> 4. We thank the reviewer for pointing this out. We have fixed the presentation issues in the revised manuscript. To clarify the notation in Equation 9: q is a learnable vector, while K and V are functions containing learnable parameters. We used lowercase letters for vectors and uppercase letters for functions (consistent with Equation 3).

---

### Review · Reviewer_dvCR · 2025-07-28

**Summary Of Contributions:**

This paper introduces LBMamba, a new locally bi-directional State Space Model (SSM) block divsed to improve the efficiency and performance of Mamba-based models in vision tasks. In LBMamba, the authors propose a lightweight local backward scan integrated within the forward selective scan of Mamba, eliminating the need for computationally expensive global backward scans. Moreover, the authors propose a thread-level bi-directional scanning operator that performs backward scans entirely in thread-private registers, minimizing memory traffic and inter-thread communication. Based on these, LBVim is introduced as a vision backbone that alternates scan directions every two layers to achieve global context modeling without additional backward sweeps. Extensive experiments show that the proposed methods can reduce computational overhead while maintaining a full receptive field. Besides, the proposed framework demonstrates superior performance-throughput trade-offs across multiple vision tasks like classical cv tasks and WSI-based tasks.

**Audience:**

Yes

**Broader Impact Concerns:**

No any broader impact concerns.

**Claims And Evidence:**

Yes

**Requested Changes:**

No other requested changes. Please refer to Weakness

**Strengths And Weaknesses:**

**Strengths**:
- An Efficient Architecture: The introduction of LBMamba, which integrates a local backward scan into Mamba’s forward pass, is a hardware-aware optimization and seems novel in the field. By avoiding a full global backward scan, the method significantly reduces computational overhead while maintaining performance.
- Strong Empirical Results: This paper demonstrates consistent improvements across multiple benchmarks (ImageNet, ADE20K, COCO, and three WSI datasets), showing superior accuracy-throughput trade-offs compared to existing Mamba-based and Transformer-based models.
- Hardware-Aware Optimization: The thread-level bi-directional scan seems good, as it minimizes memory traffic by keeping computations in registers. The related experimental results are promising and well-justified (Table 2).

**Weaknesses**:
- Limitation with Class Tokens: This paper mentioned that LBMamba underperforms when a class token is used, which is a common design in many vision models (*e.g.*, ViTs). This could restrict its applicability in architectures relying on such tokens.

- Missing Baselines: For natural image-based experiments, the primary comparisons are conducted against Vim and DeiT, but recent efficient architectures (*e.g.*, EfficientVMamba [1] and VMamba [2]) are not included. For WSI-based ones, a recent Mamba-based approach, M3amba [3], is missing. A broader comparison would better situate LBVim’s efficiency gains.

- Training Efficiency: This paper focuses on inference speed but does not provide a detailed comparison of training time or memory usage during training, which is important for large-scale applications. The authors are encouraged to discuss it.

Reference:

[1] Pei et al., EfficientVMamba: Atrous Selective Scan for Light Weight Visual Mamba, arXiv Preprint, 2024.

[2] Liu et al., VMamba: Visual State Space Model, NeurIPS, 2024.

[3] Zheng et al., M3amba: Memory Mamba is All You Need for Whole Slide Image Classification, CVPR, 2025.

---

> ### Author Response · Authors · 2025-09-01
>
> We appreciate the reviewer's valuable comment and address the issues as follows:
>
> 1. We agree with the reviewer that the class token is a current limitation of our method, and we discussed it in the limitations section. However, this limitation is not pertinent to numerous applications, such as most networks in dense prediction tasks like semantic segmentation (e.g., UperNet, Mask R-CNN) and object detection, which do not utilize a class token. In these scenarios, the linear complexity of our method is particularly advantageous for handling high-resolution inputs. We consider the extension of our method to accommodate class tokens a future work.
>
> 2. We would like to emphasize that we are not proposing a new network architecture intended to surpass the current SOTA methods, but rather an enhanced hardware-aware algorithm designed to accelerate existing architectures.  Our method facilitates the scaling of the existing models with much less computational overhead. We believe this work will inspire the community to further investigate GPU-level optimizations.
> Following the reviewer's recommendations, we applied our method to VMamba-Nano (7M parameters) and observed a performance improvement of 1.1%. To further demonstrate the generalizability of our approach, we also apply it to LocalVim and PlainMamba. As the results in Table below show, our method consistently improves the performance across these SOTA models.
> Regarding the other baselines mentioned, EfficientVMamba is a hybrid architecture of mamba and inverted residual blocks, which contains only 4 Mamba layers occupying only a small percentage of the overall architecture. Since our method specifically enhances the efficiency of Mamba layers, applying it to EfficientVMamba does not yield significant merit. As for M3amba, the official source code was not publicly available (https://github.com/titizheng/M3amba) at the time of this comment, which precluded an experimental comparison.
> We have incorporated a discussion of all these baselines into our revised manuscript.
>
> | Method | #Param. | T.P. | Top-1 acc. |
> | :--- | :---: | :---: | :---: |
> | VMamba-Nano | 7M | 998 | 78.1 |
> | LBVMamba-Nano | 9M | 1003 | **79.2** |
> | PlainMamba-L1 | 7M | 385 | 77.9 |
> | LBPlainMamba-L1 | 15M | 385 | **80.6** |
> | LocalVim-T (w.o. search) | 8M | 331 | 75.8 |
> | LocalVim-T (w. search) | 8M | 331 | 76.2 |
> | LBLocalVim-T (w.o. search) | 15M | 356 | **79.6** |
>
> 3. For a detailed analysis of training efficiency, we respectfully refer the reviewer to Appendix B in the manuscript.

---

### Review · Reviewer_TUr6 · 2025-08-04

**Summary Of Contributions:**

LBVim introduces a thread-level bidirectional state-space module combining Mamba’s linear complexity with register-optimized local backward scans. The LBVim backbone achieves full receptive fields via alternating scan directions across layers, eliminating costly global reverse passes. Key results across four vision tasks show (1) Efficiency: Just 2% runtime overhead for 83% higher throughput
(2) Accuracy: Outperforms Vim by +1.6% (ImageNet), +2.7% mIoU (ADE20K), +0.9% APb/+1.1% APm (COCO), +1.67% AUC (WSI)
(3) Scalability: Superior accuracy-throughput Pareto frontiers at scale

**Audience:**

Yes

**Broader Impact Concerns:**

WSI performance gains (↑3.06% AUC) might encourage deployment without clinical trials, risking false positives in diagnostics.

**Claims And Evidence:**

Yes

**Requested Changes:**

（1）I think the parameters, FLOPs, throughput, and accuracy should be balanced, and thus the structure of LBVim should be modified.
（2）The visulized results of Semantic Segmentation, Object Detection and Instance Segmentation should be added.

**Strengths And Weaknesses:**

Strengths:
(1) This paper proposes a novel scanning strategy for Mamba that speeds up the throughput with about 2% runtime overhead.
(2) This paper is well-organized and easy to follow.

Weaknesses:
(1) While achieving superior throughput-accuracy trade-offs, LBVim-300 demonstrates notably higher model complexity than Vim-Ti (+14% parameters, +50% FLOPs) under comparable throughput conditions.
(2) In Section 4, the authors say that "we set the number of elements each thread thread process (M ) based on the sequence length L: when L > 256 (images larger than 256 ×256), a thread processes M = 16 elements; for 128 < L ≤256 (images between 256 ×256 and 256 ×128), it processes M = 8 elements; and when L ≤128, the workload is reduced to M = 4." How is the value of M determined? More releated ablation studies should be added.
(3) it lacks visualized results for Semantic Segmentation, Object Detection, and Instance Segmentation, which could provide a clearer understanding of the model’s performance and effectiveness in these tasks.

---

> ### Author Response · Authors · 2025-09-01
>
> We appreciate the reviewer's valuable comment and address the issues as follows:
>
> 1.  We clarify that our primary objective is to achieve a trade-off between model complexity, accuracy, and efficiency. Although LBVim-300 has more parameters, the training/inference speed is not negatively affected. As discussed in section 4.1, our method enables a novel way to develop Mamba-based methods by scaling the model for higher accuracy, without slowing down the inference and training speed.
> Regarding the concern about FLOPs (i.e., the total number of floating-point operations), while they are a common efficiency metric, they operate under the assumption that all floating-point operations are computationally equivalent. Our work, however, proposes a low-level, hardware-aware algorithm that makes these operations significantly less expensive than in standard implementations. Consequently, FLOPs do not provide an accurate measure of our model's practical speed. Therefore, we present FLOPs for reference only and contend that throughput is the more representative metric of efficiency in this context.
>
> 2. The hyper-parameter $M$ represents the number of elements processed by each thread. We follow the implementation in the vanilla Mamba. Specifically, we want to maximize hardware occupancy, which requires $M \leq \mathrm{ceil}(L / 32)$. This is known as the warp occupancy maximization of GPU algorithms. $M$ also corresponds to the kernel's receptive field, for which a larger value is generally preferable. The current maximum value of $M=16$ is bounded by hardware resources, namely the number of registers available per thread.
>
> 3.  Thank you for the suggestion. We have now included visualizations in the Appendix G.

---

### Decision · Action_Editor_KVo5 · 2025-10-22

**Recommendation:** Accept with minor revision

**Additional Comments:**

The paper tackles an interesting and relevant problem, and the proposed approach has potential. However, the evaluation should be aligned with standard practices by including comparisons with models of similar parameter counts. The distinction from Adventurer (CVPR 2025) should be clarified.

**Audience:**

Yes

**Audience Explanation:**

The topic and proposed approach address efficiency and model design issues that are relevant to the TMLR audience. While the contribution is somewhat incremental, the findings could still interest readers working on lightweight or efficient architectures.

**Claims And Evidence:**

Yes

**Claims Explanation:**

The claims are generally supported but not fully convincing. The authors provide some evidence for their approach, though the reliance on FPS as a comparison metric is unconventional. Strengthening evaluations using standard metrics (e.g., models with similar parameter counts) would make the evidence clearer.